# Application of Artificial Neural Networks to Identify Alzheimer’s Disease Using Cerebral Perfusion SPECT Data

**DOI:** 10.3390/ijerph16071303

**Published:** 2019-04-11

**Authors:** Dariusz Świetlik, Jacek Białowąs

**Affiliations:** 1Intrafaculty College of Medical Informatics and Biostatistics, Medical University of Gdańsk, 1 Debinki St., 80-211 Gdańsk, Poland; 2Department of Anatomy and Neurobiology, Medical University of Gdańsk, 1 Debinki St., 80-211 Gdańsk, Poland; jacekwb@gumed.edu.pl

**Keywords:** neural networks, SPECT, Alzheimer’s disease

## Abstract

The aim of this study was to demonstrate the usefulness of artificial neural networks in Alzheimer disease diagnosis (AD) using data of brain single photon emission computed tomography (SPECT). The results were compared with discriminant analysis. The study population consisted of 132 clinically diagnosed patients. There were 72 subjects with AD and 60 belonging to the normal control group. The artificial neural network used 36 numerical values being the count numbers obtained for each area of brain SPECT. These numbers determined the set of input data for the artificial neural network. The sensitivity of Alzheimer disease diagnosis detection by artificial neural network and discriminant analysis were 93.8% and 86.1%, respectively, and the corresponding specificity was 100% and 95%. We also used receiver operating characteristic curve (ROC) analysis and areas under receiver operating characteristics curves were correspondingly 0.97 (*p* < 0.0001) for the artificial neural networks (ANN) and 0.96 (*p* < 0.0001) for discriminant analysis. In conclusion, artificial neural networks and conventional statistics methods (discriminant analysis) are a useful tool in Alzheimer disease diagnosis.

## 1. Introduction

Alzheimer’s disease is one of the most common causes of dementia in the growing population of elderly people. Despite numerous studies, both primary and strictly clinical, the pathological mechanisms of its etiology are not clear enough to permit rationally grounded prevention, early detection and effective treatment of the disease at an early stage [1]. At present, it is merely possible to delay the progression of the disease. Diagnosing Alzheimer’s disease (AD) at an early stage is hindered by the low specificity and variability of the clinical symptoms and the current lack of a biological marker with established and satisfactory diagnostic efficacy. There are attempts to use neuroimaging techniques such as functional magnetic resonance imaging (fMRI), positron emission tomography (PET) and single photon emission computed tomography (SPECT) to search for characteristic signs of early AD. We also have made some attempts to use more sophisticated artificial intelligence and neural network techniques for analysis of MRI or PET brain images [2,3,4]. Recently, researchers’ interest has been focusing on imaging techniques of metabolism of chemical compounds participating in or associated with the amyloid cascade [5]. They are usually based on the PET technique and unique markers. Although the cognitive value of these studies cannot be overestimated, no method of initial differentiation of dementia has been developed yet that would be available and cost-effective enough. Cerebral blood flow imaging by means of SPECT is a test whose usefulness has also been thoroughly studied, which is very important, with reference to the ultimate histopathological diagnosis. Though some papers have shown that the test is slightly less precise than the flow imaging by means of [^18^F] Fluorodeoxyglucose positron emission tomography (FDG-PET), the advantages of SPECT as the first-line test are its better availability and lower costs [6,7,8,9]. The purpose of this paper was an attempt to modify the cerebral blood flood evaluation by means of SPECT using artificial neural networks to raise the diagnostic usefulness of the test, making the imaging test assessment objective and, ultimately, developing a tool facilitating the detection of discreet changes.

Artificial neural networks (ANN) inspired by the biological model are patterned on brain functioning [10,11,12,13]. Diagnostic decision support, analysis and interpretation of images are the medical issues solved by means of artificial intelligence methods, the ANN in particular [14,15,16,17,18,19]. Artificial neural networks are used in contemporary nuclear medicine to support the diagnostics of ischemic heart disease [20,21,22], pulmonary embolism [23,24], parathyroid adenoma [24], Alzheimer’s disease [25,26,27,28,29,30] and breast cancer [31]. Apart from those, artificial neural networks are useful tools in dentistry for predicting the number of follow-up visit [32]. 

Alzheimer’s disease is responsible for 50–70% of causes of dementia in Europe. Epidemiological studies indicate that AD incidence rises with age—it is diagnosed in approximately 14% of patients over 65 and in approximately 40% of patients over 80. 

Our study describes the application of artificial neural networks to the diagnostics of Alzheimer’s disease on the basis of information contained in digital images of SPECT cerebral blood flow assessments.

The indicators of an efficient study are sensitivity and specificity of the examination methods. The sensitivity expresses a probability of the correct recognition of a disease with the use of a given diagnostic test. The specificity expresses a probability of the correct exclusion of the opposite/not affected/injured cases with the use of a given diagnostic test. The receiver operating characteristic curve (ROC) shows a relationship between sensitivity and the “specificity supplement to 1”. An area under the curve (interval of values: 0 to 1) reflects an ability of the test to properly segregate true and false outcomes. In addition, on this basis the segregation ability of various tests can be compared. 

## 2. Materials and Methods

### 2.1. Patients’ Population

The study brain SPECT was performed in the Department of Nuclear Medicine in the Medical University of Gdańsk (Poland). The study population consisted of 132 clinically diagnosed patients. 72 (43 female, 29 male) had AD and 60 (44 female, 16 male) were the normal control group. Age range in the Alzheimer group was from 55 to 87 years (mean (standard deviation (SD)) 69.7 (10.0), and in the normal group was from 54 to 82 years with a mean (SD) of 64.9 (9.8). The brain SPECT study were evaluated for each patient. AD was diagnosed based on the diagnostic criteria of dementia of the American Psychiatric Association (DSM-IV) [33]. In all patients magnetic resonance imaging (MRI, *n* = 47) or a computerized tomography study (CT, *n* = 3, due to contraindications to perform MRI) was performed to exclude alternative causes of dementia, like a tumor, stroke or hydrocephalus.

### 2.2. Cerebral Perfusion SPECT

SPECT cerebral blood flow testing was performed at the Medical University of Gdańsk Institute of Radiology and Nuclear Medicine Department of Nuclear Medicine in 2000–2005. The tomographic study of cerebral blood flow was performed 20 min after intravenous administration of Tc-99m-HMPAO (Amersham, United Kingdom) with an activity of 20 mCi (740 MBq), using a triple head gamma camera Multispect-3 (Siemens, Erlangen, Germany), equipped with a special collimator for neurological tests (Neurofocal). Data were collected in a 128 × 128 matrix during rotation along a 120 arch. Data acquisition and reconstruction was conducted using an ICON computer (Siemens, Erlangen, Germany). Data reconstruction was performed using a Butterworth filter, cut-off frequency 0.35, layer thickness 2 pixels (9.6 mm).

### 2.3. Input Signals for Artificial Neural Network

Artificial neural network used information from image records SPECT study. We prepared for each patient a set of 36 numerical values. These values each corresponded a particular brain area to a brain profile (Figure 1). All brain profiles (Parietal, Ventricular, and Thalamus) contained 12 areas. These variables were the input signals for the neural network.

### 2.4. The Architecture of the Artificial Neural Network

We used a multilayer perceptron network in our study. The artificial neural network consisted of three layers: The input, hidden and output layers of 36, 21 and 1 neurons, respectively. Using a software simulator of artificial neural networks we created networks to solve regression tasks which consisted of predicting the number of corrections. The answer of the artificial neural network to each test case fell within a numerical range of 0 to 1. The activation and rejection levels for the output neuron were selected automatically by the stimulator of artificial neural networks in order to minimize the losses. During the learning process the weight links between the neurons were modified using the error back-propagation algorithm. The sum of squared differences between the a priori given values and the actual values at the output neuron was chosen as the error function for the artificial neural network. A sigmoid (logistic function) was used as the activation function. The learning coefficient was 0.01 and the inertia was 0.3. The number of 1000 epochs was established, where the order of presented cases for the neural network was different in each epoch. The initialization of neural network weights was done by random Gaussian method.

To calculate the diagnostic quality of the artificial neural network, we divided at random of all patients for two groups: Training and testing. The training group contained 100 (55 AD, 45 normal) and testing group 32 (17 AD, 15 normal) cases. 

### 2.5. The Software Simulation of ANN and the Statistical Analysis

All calculations were performed with the use of the software emulator of artificial neural networks TIBCO Software Inc., Statistica (data analysis software system), version 13 (Palo alto, CA, USA, 2017, http://statistica.io). Sensitivity and specificity as well as the ROC analysis of the Alzheimer disease diagnosis and discriminant analysis were estimated using TIBCO Software Inc. (2017). Performance of the neural networks and discriminant analysis were evaluated by ROC curve. The area under the ROC curve was used as the performance of the artificial neural network and discriminant analysis. We used a discriminant analysis classifier, like Kippenhan et al. [28,29] who used this type of classifier in the evaluation of a neural network classifier applied to perfusion profiles extracted from PET scans. For our simulations presented here, we used the L-method of cross-validation testing. Discriminant analysis is very similar to the analysis of variance from a conceptual and computational point of view. In this statistical procedure, variables (SPECT counts) were being searched for, whose average values were different in the examined groups (AD group and control), and which importantly allow prediction of the correct assignment of new cases to the groups. Discriminant analysis was carried out in two stages. In the first one, discriminant functions were created, which are a linear combination of input variables, in order to best represent the variance of the data set and reduce the dimensionality of the data set. The procedure led to the maximization of the difference in mean values of discriminatory functions in particular subgroups. In the second stage, classification functions were created, which are also linear combinations of independent variables. These functions determined the probability of a given case belonging to one of the studied categories: AD or control group. As in the case of neural networks, in the discriminant analysis we utilized the following division of the entire study group: Training group (55 AD, 45 normal) and testing group (17 AD, 15 normal). The discriminant analysis classifier gave an ROC area. Six sections from each area of brain (parietal, ventricular, thalamus) were analyzed, generating six count numbers in the AD group and the control. Statistical analysis, both one-way and two-way ANOVA, was performed using Statistica software (Dell, Round Rock, TX, USA). The accepted significance level was *p* < 0.05. 

## 3. Results

A statistical analysis showed significant differences in mean scintillation signals (SPECT counts) in particular areas of the brain between AD patients and the control group Table 1, Table 2 and Table 3. 

The two-way ANOVA combined analysis of six zones of the parietal area (prefrontal high, medial frontal high, central high, parietal sup. (superior) A, parietal sup. B and parietal sup. C) showed a significant decrease in the count number of AD group as compared to the control (*p* < 0.0001). No significant effect of the side of the brain on the count number (SPECT counts) of the parietal area was found (*p* > 0.05).

The two-way ANOVA combined analysis of six zones of the parietal area (prefrontal mid, medial frontal low, central low, parietal inf. (inferior), parieto-occipital and occipital sup.) showed a significant decrease in the count number of AD group as compared to the control (*p* < 0.0001). No significant effect of the side of the brain on the count number (SPECT counts) of the parietal area was found (*p* > 0.05).

The two-way ANOVA combined analysis of six zones of the parietal area (prefrontal low, infero-frontal, basal ggl. (ganglia), thalamus, temporal sup. and occipital inf.) showed a significant decrease in the count number of AD group as compared to the control (*p* < 0.0001). No significant effect of a side of the brain on the count number (SPECT counts) of the parietal area was found (*p* > 0.05).

SPECT cerebral brain flow tests demonstrated that AD patients had significantly lower values of scintillation calculations in all the areas of the brain in comparison to the control group. The thalamus section was characterized by the lowest mean value of scintillation calculations for both study groups in both cerebral hemispheres (Figure 1). 

The AD diagnostic sensitivity of artificial neural networks was 93.8% (standard deviation 4.3%) and the specificity was 100% (0.1%). The sensitivity obtained in the learning group was 98.1% (7.4%), and the specificity was 100% (0.1%). In the discriminatory analysis they were 86.1% (6.9%) and 95% (5.7%), respectively. 

For a more precise analysis, ROC curves were crossed out and areas under the curves were calculated for ANN (0.97, *p* < 0.0001) and discriminatory analysis (0.96, *p* < 0.0001). ROC curves provided an estimation of AD diagnostic power of an artificial neural network and the traditional statistical method (discriminatory analysis) (Figure 2).

## 4. Discussion

Several studies have reported that detecting occipital and parietotemporal hypoperfusion could be useful in differentiation of AD from dementia with Lewy bodies (DLB) [33]. However, hypoperfusion in the parietotemporal lobe has been observed in the brains of patients with both DLB and frontotemporal dementia and brain perfusion SPECT in some demented disorders can be similar to that in AD [34]. Dougall et al. suggested that perfusion SPECT is unable to draw a clear line between AD and other dementias [35]. Nevertheless, SPECT may still be helpful in the clinical differential diagnosis of AD if used in conjunction with other neuroimaging techniques [35].

An artificial neural network is a tool that, unlike traditional statistical methods, uses the learning process in the cases described. In our study, ANN used as its input the information from a SPECT cerebral blood flow test from 132 patients. For each patient a set of 36 numbers was prepared, representing the scintillation calculation in particular areas of the brain. A comparison of AD diagnosis by means of ANN and discriminatory analysis has not shown any statistically significant differences (chi-square 0.7, *p* = 0.4).

The created neural network was more effective at differentiating between patients with AD and healthy subjects in comparison with other tests where the value of the area under the ROC curve was 0.91 and 0.93 [27,29]. Furthermore, Chan et al. constructed a neural network with 120 inputs, using information from perfusion cerebral flow imaging in 81 patients [24]. 

The number of neural network inputs is highly diversified in the literature to date. The smallest number of ANN inputs is four, while each of them constituted the mean value of scintillation calculations [26]. The highest number of inputs was 120, used by Chan et al., each corresponding to a standardized cortical region [24]. 

The effectiveness of AD diagnosing by means of discriminatory analysis was also greater in our study than in similar papers: 0.85 and 0.94 [26,27]. 

Moreover, our effects have been presented as sensitivity and specificity of AD diagnosis by means of ANN and discriminatory analysis. Sensitivity defines the probability of correct AD diagnosis and specificity defines the probability of correct exclusion of the disease.

### Limitation

Highly similar values of the ROC area obtained for ANN and discriminatory analysis (0.97 and 0.96) as well as no statistically significant difference in the sensitivity (93.8% and 86.1%) did not support any significant difference between the proposed method and the discriminatory analysis. In addition to that, there are obviously a number of classification methods which have not been considered by authors.

## 5. Conclusions

Artificial neural networks and conventional statistics methods (discriminant analysis) are a useful tool in Alzheimer disease diagnosis. The results of our study indicate that artificial neural networks have the capacity to discriminate AD patients from healthy controls. Our study simulations provide evidence that artificial neural networks can be a useful tool for clinical practice.

## Figures and Tables

**Figure 1 ijerph-16-01303-f001:**
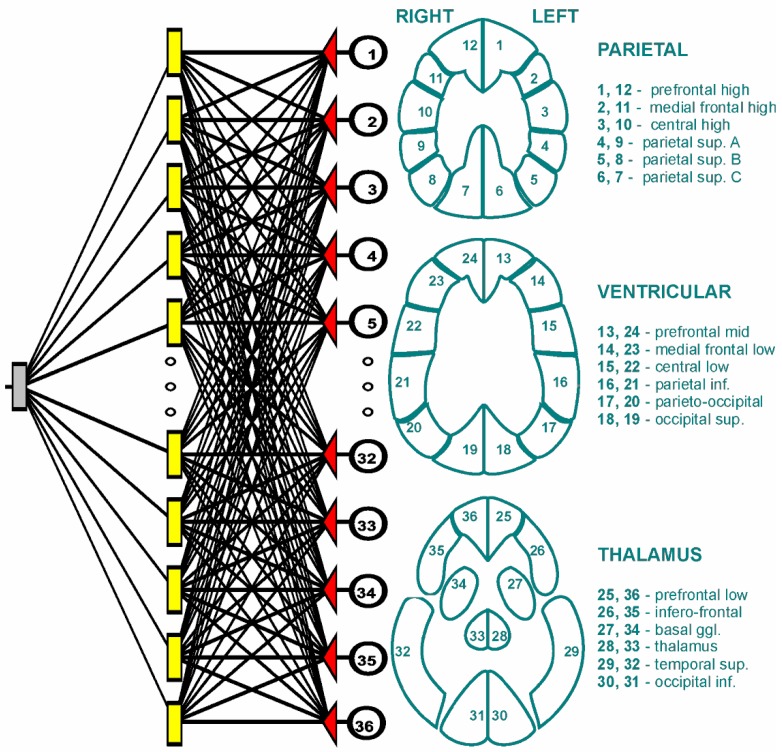
On the left panel is shown the architecture of the artificial neural network. On the right panel are presented the brain profiles that each contain 12 areas.

**Figure 2 ijerph-16-01303-f002:**
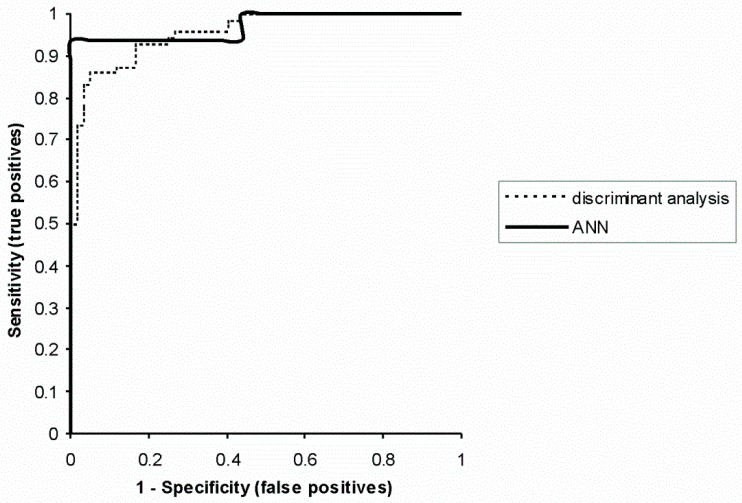
Receiver operating characteristic (ROC) curves for the artificial neural network (ANN) and discriminatory analysis.

**Table 1 ijerph-16-01303-t001:** Comparison count number parietal area of the brain for Alzheimer’s disease (AD) patients and control group (mean and SD).

Parietal Area	Right		*p*-Value	Left		*p*-Value
	AD	Control		AD	Control	
Prefrontal high (1, 12)			0.0001			0.0002
Mean	125.22	176.48		124.63	174.03	
(SD)	(51.32)	(91.46)		(51.74)	(88.88)	
Medial frontal high (2, 11)			0.0001			0.0001
Mean	125.90	182.15		127.57	181.97	
(SD)	(55.10)	(92.11)		(53.11)	(92.39)	
Central high (3, 10)			0.0008			0.0003
Mean	129.64	181.67		128.60	180.83	
(SD)	(56.22)	(98.15)		(54.01)	(94.42)	
Parietal sup. A (4, 9)			0.0001			0.0001
Mean	126.14	181.17		124.60	178.95	
(SD)	(55.26)	(95.57)		(54.03)	(92.78)	
Parietal sup. B (5, 8)			0.0001			0.0001
Mean	124.14	183.60		123.72	182.32	
(SD)	(55.44)	(93.36)		(55.27)	(92.37)	
Parietal sup. C (6, 7)			0.0002			0.0002
Mean	130.51	186.12		131.49	185.23	
(SD)	(55.28)	(97.46)		(55.27)	(95.80)	

sup.: superior.

**Table 2 ijerph-16-01303-t002:** Comparison count number ventricular area of the brain for AD patients and control group (mean and SD).

Ventricular Area	Right		*p*-Value	Left		*p*-Value
	AD	Control		AD	Control	
Prefrontal mid (13, 24)			0.0001			0.0003
Mean	125.43	176.27		125.88	174.80	
(SD)	(51.33)	(87.71)		(51.92)	(88.43)	
Medial frontal low (14, 23)			0.0001			0.0001
Mean	123.79	178.27		124.94	178.97	
(SD)	(52.49)	(92.50)		(51.65)	(90.32)	
Central low (15, 22)			0.0007			0.0003
Mean	124.76	171.87		122.81	172.03	
(SD)	(52.13)	(91.69)		(50.97)	(90.58)	
Parietal inf. (16, 21)			0.0001			0.0001
Mean	123.89	178.63		121.54	176.18	
(SD)	(53.41)	(94.02)		(51.19)	(92.58)	
Parieto-occipital (17, 20)			0.0001			0.0001
Mean	126.26	180.22		124.10	180.15	
(SD)	(55.05)	(88.28)		(54.13)	(90.78)	
Occipital sup. (18, 19)			0.0008			0.0011
Mean	137.49	189.67		137.04	188.97	
(SD)	(58.07)	(97.51)		(57.45)	(97.65)	

inf.: inferior; sup.: superior.

**Table 3 ijerph-16-01303-t003:** Comparison count number thalamus area of the brain for AD patients and control group (mean and SD).

Thalamus Area	Right		*p*-Value	Left		*p*-Value
	AD	Control		AD	Control	
Prefrontal low (25, 36)			0.0011			0.0011
Mean	61.43	84.52		60.89	84.42	
(SD)	(26.29)	(44.54)		(25.85)	(44.45)	
Infero-frontal (26, 35)			0.0002			0.0002
Mean	63.04	89.22		62.86	89.13	
(SD)	(26.51)	(46.34)		(26.03)	(47.30)	
Basal ggl. (27, 34)			0.0009			0.0008
Mean	64.63	88.87		64.69	89.20	
(SD)	(27.62)	(47.71)		(26.71)	(47.65)	
Thalamus (28, 33)			0.0011			0.0019
Mean	67.76	91.82		67.68	91.67	
(SD)	(28.76)	(47.21)		(29.01)	(46.44)	
Temporal sup. (29, 32)			0.0061			0.0053
Mean	75.08	100.48		74.56	100.72	
(SD)	(35.66)	(54.36)		(36.20)	(55.52)	
Occipital inf. (30, 31)			0.0024			0.0020
Mean	65.68	88.00		65.00	88.60	
(SD)	(27.84)	(45.39)		(27.90)	(45.71)	

inf.: inferior; sup.: superior; ggl.: ganglia.

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
