# Peer review of "Application of Artificial Neural Networks to Identify Alzheimer’s Disease Using Cerebral Perfusion SPECT Data"

_ijerph, 2019, doi:10.3390/ijerph16071303_

Reviewer 1 Report

Authors addressed all the proposed changes successfully. 

Author Response

English language and style are fine/minor spell check required

1.   Completion was added in red in lines 106-108:

The sum of squared differences between the a priori given values and the actual values at the output neuron was chosen as the error function for the artificial neural network.

2.   Completion was added in red in lines 124-139:

Discriminant analysis is very similar to the analysis of variance from a conceptual and computational point of view.  In this statistical procedure, variables (SPECT counts) are being searched for, whose average values are different in the examined groups (AD group and control), and which importantly allow to predict the correct assignment of new cases to the groups. Discriminant analysis is carried out in two stages. In the first one, discriminant functions are created, which are a linear combination of input variables, in order to best represent the variance of the data set and reduce the dimensionality of the data set. The procedure leads to the maximization of the difference in mean values of discriminatory functions in particular subgroups.  In the second stage, classification functions are created, which are also linear combinations of independent variables. These functions determine the probability of a given case belonging to one of the studied categories: AD or control group. As in the case of neural networks, in the discriminant analysis we utilized the following division of the entire study group: training group (55AD, 45 normal) and testing group (17 AD, 15 normal). The discriminant analysis classifier gave an ROC area. From each area of brain (parietal, ventricular, thalamus) six section were analyzed generating six count number in AD group and control. Statistical analysis, both one-way and two-way ANOVA, was performed using Statistica software.

3.   Completion was added in red in lines 144-155:

The two-way ANOVA combined analysis of six zones of the parietal area (prefrontal high, medial frontal high, central high, parietal sup. A, parietal sup. B and parietal sup. C) showed a significant decrease in the count number of  AD group as compared to the control (p<0.0001). no="" significant="" effect="" of="" a="" side="" the="" brain="" on="" count="" number="" spect="" parietal="" area="" was="" found="" p="">0.05).

The two-way ANOVA combined analysis of six zones of the parietal area (prefrontal mid, medial frontal low, central low, parietal inf., parieto-occipital and occipital sup.) showed a significant decrease in the count number of  AD group as compared to the control (p<0.0001). no="" significant="" effect="" of="" a="" side="" the="" brain="" on="" count="" number="" spect="" parietal="" area="" was="" found="" p="">0.05).

The two-way ANOVA combined analysis of six zones of the parietal area (prefrontal low, infero-frontal, basal ggl., thalamus, temporal sup. and occipital inf.) showed a significant decrease in the count number of  AD group as compared to the control (p<0.0001). no="" significant="" effect="" of="" a="" side="" the="" brain="" on="" count="" number="" spect="" parietal="" area="" was="" found="" p="">0.05).

4.   Completion was added in red in lines 216-221:

4.1. Limitation

Highly similar values of the ROC area obtained for ANN and discriminatory analysis (0.97 and 0.96) as well as no statistically significant difference in the sensitivity (93.8% and 86.1%) did not support any significant difference between the proposed method and the discriminatory analysis. In addition to that, there are obviously a number of classification methods which have not been considered by authors.

Reviewer 2 Report

In this manuscript, the authors claim to describe the application of artificial neural networks to the diagnostics of Alzheimer’s disease on the basis of information contained in digital images of SPECT cerebral blood flow assessments. 

Although some critical points have been clarified and the manuscript is more understandable, there are still some points to be revised:

1. no answer has been given to the request to carry out statistical correction on multiple comparisons. 

2. The authors should better explain the functioning of the discriminant analysis classifier that they use in comparison with the proposed ANN method.

3. As reported in section "Discussion" : 

"highly similar values of the ROC area for ANN and discriminatory analysis (0.97 and 0.96) and no statistically significant difference in sensitivity"

they did not find significant differences between the proposed method and the discriminatory analysis. In addition, there are a number of classification methods that have not been considered by the authors.

It is necessary to highlight these limitations in discussions or in a new section "Limitation" section.

Author Response

1. no answer has been given to the request to carry out statistical correction on multiple comparisons.

Completion was added in red in lines 144-155:

The two-way ANOVA combined analysis of six zones of the parietal area (prefrontal high, medial frontal high, central high, parietal sup. A, parietal sup. B and parietal sup. C) showed a significant decrease in the count number of  AD group as compared to the control (p<0.0001). no="" significant="" effect="" of="" a="" side="" the="" brain="" on="" count="" number="" spect="" parietal="" area="" was="" found="" p="">0.05).

The two-way ANOVA combined analysis of six zones of the parietal area (prefrontal mid, medial frontal low, central low, parietal inf., parieto-occipital and occipital sup.) showed a significant decrease in the count number of  AD group as compared to the control (p<0.0001). no="" significant="" effect="" of="" a="" side="" the="" brain="" on="" count="" number="" spect="" parietal="" area="" was="" found="" p="">0.05).

The two-way ANOVA combined analysis of six zones of the parietal area (prefrontal low, infero-frontal, basal ggl., thalamus, temporal sup. and occipital inf.) showed a significant decrease in the count number of  AD group as compared to the control (p<0.0001). no="" significant="" effect="" of="" a="" side="" the="" brain="" on="" count="" number="" spect="" parietal="" area="" was="" found="" p="">0.05).

2. The authors should better explain the functioning of the discriminant analysis classifier that they use in comparison with the proposed ANN method.

Completion was added in red in lines 124-139:

Discriminant analysis is very similar to the analysis of variance from a conceptual and computational point of view.  In this statistical procedure, variables (SPECT counts) are being searched for, whose average values are different in the examined groups (AD group and control), and which importantly allow to predict the correct assignment of new cases to the groups. Discriminant analysis is carried out in two stages. In the first one, discriminant functions are created, which are a linear combination of input variables, in order to best represent the variance of the data set and reduce the dimensionality of the data set. The procedure leads to the maximization of the difference in mean values of discriminatory functions in particular subgroups.  In the second stage, classification functions are created, which are also linear combinations of independent variables. These functions determine the probability of a given case belonging to one of the studied categories: AD or control group. As in the case of neural networks, in the discriminant analysis we utilized the following division of the entire study group: training group (55AD, 45 normal) and testing group (17 AD, 15 normal). The discriminant analysis classifier gave an ROC area. From each area of brain (parietal, ventricular, thalamus) six section were analyzed generating six count number in AD group and control.

3. As reported in section "Discussion" :

"highly similar values of the ROC area for ANN and discriminatory analysis (0.97 and 0.96) and no statistically significant difference in sensitivity"

they did not find significant differences between the proposed method and the discriminatory analysis. In addition, there are a number of classification methods that have not been considered by the authors.

It is necessary to highlight these limitations in discussions or in a new section "Limitation" section.

The Limitation section was added and completed in red according to the recommendations.

4.1. Limitation

Highly similar values of the ROC area obtained for ANN and discriminatory analysis (0.97 and 0.96) as well as no statistically significant difference in the sensitivity (93.8% and 86.1%) did not support any significant difference between the proposed method and the discriminatory analysis. In addition to that, there are obviously a number of classification methods which have not been considered by authors.

Moderate English changes required

It was be done

This manuscript is a resubmission of an earlier submission. The following is a list of the peer review reports and author responses from that submission.

Round  1

Reviewer 1 Report

1 Need to increase network selection and construction work

2 Multimodal image comparison experiments are required

3 English needs to be strengthened

Author Response

We fully agree with reviewer suggestions and the second part of manuscript was excluded for further separate publication.

1 Need to increase network selection and construction work

It was done. The aim was to present quite different methods of use computer aid analysis of Alzheimer Disease, which could put forward our knowledge of them. In reorganized manuscript  we send for publication only the first part about the ANN analysis of SPECT investigations. The modeling of Brain hippocampal microcircuits will be the matter of further separate publication

2 Multimodal image comparison experiments are required

In the present study images were not analysed. But rather instead of visual analysis we were focused on quantitative numbers.

3 English needs to be strengthened

It was be done

Reviewer 2 Report

In this manuscript authors presents a classification method based on artificial neural networks to detect Alzheimer’s disease using brain SPECT data from 132 different diagnosed patients. The results are quite promising, achieving high sensitivity and specificity results. Since this reviewer is not an expert in the understanding of the methods employed to perform the simulations CA3-CA1, this reviewer does not feel qualified to evaluate these parts. On the other hand, regarding to the computation of the evaluation metrics for the comparison of the classification methods (ANN vs. Discriminatory Analysis) there are some minor questions that should be clarified within the manuscript:

Which method was employed to compute the classification results of sensitivity and specificity? Leave-one-patient-out cross-validation, K-fold cross-validation…

Could the authors provide the standard deviation of the sensitivity and specificity results for each classifier?

Could the authors provide an estimation of the time required to compute the classification result of one patient?

General comments:

Extensive English editing should be performed throughout the manuscript.

Several acronyms are not defined in the manuscript, such as MRI, PET, FDG-PET, MS, EC, LM, EPSPd, IPSPd, AMPA, NMDA, GABA-A, LTP.

Maybe a brief description of the CA1 and CA3 hippocampus regions should be included in the introduction for non-expert readers.

Line 93 and 94: What is the meaning of the numbers between brackets?

Line 121: Please, rephrase the sentence: “As the error function for the artificial neural network was chosen the sum of squared differences between the a priori given values and the actual values at the output neuron.”

Line 228: Please, rephrase the sentence: “In the theta phase separation of these processes in half-cycles of theta a basic role in played by dendric, somatic and axonal inhibition [56-59].”

Conclusions should be a little bit more extended, summarizing the main outcomes of the manuscript.

Author Response

We fully agree with reviewer suggestions and the second part of manuscript was excluded for further separate publication.

In this manuscript authors presents a classification method based on artificial neural networks to detect Alzheimer’s disease using brain SPECT data from 132 different diagnosed patients. The results are quite promising, achieving high sensitivity and specificity results. Since this reviewer is not an expert in the understanding of the methods employed to perform the simulations CA3-CA1, this reviewer does not feel qualified to evaluate these parts. On the other hand, regarding to the computation of the evaluation metrics for the comparison of the classification methods (ANN vs. Discriminatory Analysis) there are some minor questions that should be clarified within the manuscript:

Which method was employed to compute the classification results of sensitivity and specificity? Leave-one-patient-out cross-validation, K-fold cross-validation…

To calculate the diagnostic quality of the artificial neural network we divided at random of all patients for two groups: training and testing. The training group contained 100 (55 AD, 45 normal) and testing group 32 (17 AD, 15 normal) cases.

Patient group compositions for discriminating between AD and normal controls

Training group

55 AD

45 normal

Testing group

17 AD

15 normal

Performance of the neural networks and discriminant analysis were evaluated by ROC curve. The ROC curve was constructed by selecting 10 thresholds for the output units of the network, which indicated the likelihood of being classified as AD pattern on a scale of 0 to 1. The area under the ROC curve was used as the performance of the artificial neural network. The neural network was trained to discriminate patients with AD and normal. The neural networks classifier gave an ROC area.

We used a discriminant analysis classifier like Kippenhan et al. [1,2] used this type of classifier in the evaluation of a neural-network classifier applied to perfusion profiles extracted from PET scans. For our simulations presented here, we used the L-method of cross-validation testing. The discriminant analysis classifier gave an ROC area.

1.    Kippenhan is, Barker WW, Pascal S. Nagel J, Duara R. Evaluation of a neural

network classifier for PET scans of normal and Alzheimer's disease subjects. J Nuc

Med l992;33:l459I467.

2.    Kippenhan, Barker WW, Nagel I, Grady C, Duara R. Neural-network classification

of normal and Alzheimer's disease subjects using high-resolution and low-resolution

PETcameras.J Nuc Med 1994;35:71

Could the authors provide the standard deviation of the sensitivity and specificity results for each classifier?

It was done

Could the authors provide an estimation of the time required to compute the classification result of one patient?

About half an hour.

Extensive English editing should be performed throughout the manuscript.

It was done

Several acronyms are not defined in the manuscript, such as MRI, PET, FDG-PET, MS, EC, LM, EPSPd, IPSPd, AMPA, NMDA, GABA-A, LTP.

MRI-magnetic resonance imaging

PET-positron emission tomography

FDG-PET -[18F]Fluorodeoxyglucose (FDG) positron emission tomography

MC-Mossy fibers

EC-entorinary cortex

LM- lacunosum-moleculare

EPSPd-excitatory postsynaptic potential

IPSPd-inhibory post-synaptic potential

AMPA-α-amino-3-hydroxy-5-methyl-4-isoxazolepropionic acid receptor

NMDA-N-methyl-D-aspartate receptor

GABA-A- receptors that respond to the neurotransmitter gamma-aminobutyric acid

LTP- long-term potentiation

Maybe a brief description of the CA1 and CA3 hippocampus regions should be included in the introduction for non-expert readers.

The aim was to present quite different methods of use computer aid analysis of Alzheimer Disease, which could put forward our knowledge of them. In reorganized manuscript  we send for publication only the first part about the ANN analysis of SPECT investigations. The modeling of Brain hippocampal microcircuits will be the matter of further separate publication

Line 93 and 94: What is the meaning of the numbers between brackets?

The standard deviation.

Line 121: Please, rephrase the sentence: “As the error function for the artificial neural network was chosen the sum of squared differences between the a priori given values and the actual values at the output neuron.”

The sum of squared differences between the a priori given values and the actual values at the output neuron was chosen as the error function for the artificial neural network.

Line 228: Please, rephrase the sentence: “In the theta phase separation of these processes in half-cycles of theta a basic role in played by dendric, somatic and axonal inhibition [56-59].”

This part of manuscript will be published later, separately.

Conclusions should be a little bit more extended, summarizing the main outcomes of the manuscript.

Reviewer 3 Report

In this manuscript, the authors claim to describe the application of artificial neural networks to the diagnostics of Alzheimer’s disease on the basis of information contained in digital images of SPECT cerebral blood flow assessments. They also specify in Introduction that the aim of the paper is to present computational modeling of learning and memory of the subregions CA3-CA1 of the hippocampal formation microcircuit. 

Frankly, the two objectives are not well described throughout the entire manuscript and they seem to be two parts totally separate. 

There are some points that require a complete review of the work:

1. In subsection  "2.1.3. Input signals for artificial neural network" the classification architecture is not well explained: why the authors do not use classical brain templates such as AAL or others? What are the input features? It is not very clear: what they mean with the sentence "The answer of the artificial neural network to each test case fell within a numerical range of 0 to 1" ?

2. In subsection "3.1. Artificila neural networks to detection AD", what is the "discriminatory analysis"? What kind of classifier is adopted? Did the authors consider a method for multiple testing correction such as Bonferroni or FWER? In fact, they report significant values for all the regions examined: since they carried out multiple statistical hypotheses, it is necessary to correct p-values for multiple comparisons.

3. In subsection 3.2 the authors introduced the Visual Recurrence Analysis (VRA) as a nonlinear method.  The method is neither introduced before section  "Results" nor well explained: why did the authors introduce a non-linear analysis method? What are the main objectives of this analysis? 

4. the conclusions are very short: please argue more in details the main findings of the work.

5. Bibliography refers to not recent articles: updating to more recent works is required to accurately refer to the state of the art.

In conclusion, I suggest to completely reorganise the manuscript by explaining  all the analysis procedures in section "Methods" and explain how the two elements "Medical Imaging Data" and "Simulations CA3-CA1 Hippocampal Formation Microcircuit" are used together for the common purpose of classifying the population affected by Alzheimer's Disease.

Author Response

We fully agree with reviewer suggestions and the second part of manuscript was excluded for further separate publication.

In this manuscript, the authors claim to describe the application of artificial neural networks to the diagnostics of Alzheimer’s disease on the basis of information contained in digital images of SPECT cerebral blood flow assessments. They also specify in Introduction that the aim of the paper is to present computational modeling of learning and memory of the subregions CA3-CA1 of the hippocampal formation microcircuit.

Frankly, the two objectives are not well described throughout the entire manuscript and they seem to be two parts totally separate.

We agree with this statements, the manuscript will be fully reorganized. The aim was to present quite different methods of use computer aid analysis of Alzheimer Disease, which could put forward our knowledge of them.  

1. In subsection  "2.1.3. Input signals for artificial neural network" the classification architecture is not well explained: why the authors do not use classical brain templates such as AAL or others? What are the input features? It is not very clear: what they mean with the sentence "The answer of the artificial neural network to each test case fell within a numerical range of 0 to 1" ?

To calculate the diagnostic quality of the artificial neural network we divided at random of all patients for two groups: training and testing. The training group contained 100 (55 AD, 45 normal) and testing group 32 (17 AD, 15 normal) cases.

Patient group compositions for discriminating between AD and normal controls

Training group

55 AD

45 normal

Testing group

17 AD

15 normal

Performance of the neural networks and discriminant analysis were evaluated by ROC curve. The ROC curve was constructed by selecting 10 thresholds for the output units of the network, which indicated the likelihood of being classified as AD pattern on a scale of 0 to 1. The area under the ROC curve was used as the performance of the artificial neural network. The neural network was trained to discriminate patients with AD and normal. The neural networks classifier gave an ROC area.

2. In subsection "3.1. Artificila neural networks to detection AD", what is the "discriminatory analysis"? What kind of classifier is adopted? Did the authors consider a method for multiple testing correction such as Bonferroni or FWER? In fact, they report significant values for all the regions examined: since they carried out multiple statistical hypotheses, it is necessary to correct p-values for multiple comparisons.

Artificial neural network used information from image records SPECT study. We prepared for each patients a set of 36 numerical values. They were count number in each of brain area correspondingly to brain profile. These variables were the input signals for the neural network.

We used a discriminant analysis classifier like Kippenhan et al. [1,2] used this type of classifier in the evaluation of a neural-network classifier applied to perfusion profiles extracted from PET scans. For our simulations presented here, we used the L-method of cross-validation testing. The discriminant analysis classifier gave an ROC area.

1.    Kippenhan is, Barker WW, Pascal S. Nagel J, Duara R. Evaluation of a neural

network classifier for PET scans of normal and Alzheimer's disease subjects. J Nuc

Med l992;33:l459I467.

2.    Kippenhan, Barker WW, Nagel I, Grady C, Duara R. Neural-network classification

of normal and Alzheimer's disease subjects using high-resolution and low-resolution

PETcameras.J Nuc Med 1994;35:71

3. In subsection 3.2 the authors introduced the Visual Recurrence Analysis (VRA) as a nonlinear method.  The method is neither introduced before section  "Results" nor well explained: why did the authors introduce a non-linear analysis method? What are the main objectives of this analysis?

In reorganized manuscript  we send for publication only the first part about the ANN analysis of SPECT investigations. The modeling of Brain hippocampal microcircuits will be the matter of further separate publication.

4. the conclusions are very short: please argue more in details the main findings of the work.

It was improved

5. Bibliography refers to not recent articles: updating to more recent works is required to accurately refer to the state of the art.

We have added mote that ten new references

In conclusion, I suggest to completely reorganise the manuscript by explaining  all the analysis procedures in section "Methods" and explain how the two elements "Medical Imaging Data" and "Simulations CA3-CA1 Hippocampal Formation Microcircuit" are used together for the common purpose of classifying the population affected by Alzheimer's Disease.

As we have mentioned above, we fully agree with reviewer suggestions and the second part of manuscript was excluded for further separate publication.